# Comparison of abortion incidence estimates derived from direct survey questions versus the list experiment among women in Ohio

**Robert B. Hood**[1], **Heidi Moseson**[2], **Mikaela Smith**[3], **Payal Chakraborty**[3], **Alison H. Norris**[3], **Maria F. Gallo**[3]*

**1** Department of Epidemiology, Rollins School of Public Health, Emory University, Atlanta, GA, United States of America, **2** Ibis Reproductive Health, Oakland, CA, United States of America, **3** Division of Epidemiology, College of Public Health, The Ohio State University, Columbus, OH, United States of America

* gallo.86@osu.edu

**Data Availability Statement:** The authors cannot share the data publicly because they do not own them. Data are available from NORC (via

## Abstract

Abortion is highly stigmatized in the United States which prevents its accurate measurement in surveys. The list experiment aims to improve the reporting of abortion history. We evaluated whether a list experiment resulted in higher reporting of abortion experiences than did two direct questions. Utilizing data from a representative survey of adult women of reproductive age in Ohio, we examined abortion history using two direct questions and a double list experiment. Through the double list experiment, we asked respondents to report how many of two lists of health items they had experienced; one list included abortion. We compared weighted history of abortion between these measures and by respondent demographic characteristics (age and socioeconomic status). Estimates of abortion history were similar between direct and list experiment questions. When measured with the two different direct question of abortion history, 8.4% and 8.0% of all respondents indicated ever having an abortion and with the list experiment, 8.5% indicated ever having an abortion. In a Midwestern state-level survey, the list experiment did not lead to increases in abortion reporting as compared to the direct questions. Subgroup analyses require larger samples, and future iterations should incorporate related but non-stigmatized control items to reduce misclassification and under-powering of such subgroup analyses.

## Introduction

In the United States (US), abortion is legal but increasingly restricted [1] and highly stigmatized [2–7]. As a result of stigma and privacy concerns, abortion experiences are often underreported in surveys—especially when respondents are asked directly about abortion history [8]. Having valid estimates of lifetime cumulative incidence of abortion is important to normalize utilization of this common [9] health service and inform the development of evidence-based policies to improve access to reproductive healthcare. Without valid estimates of the magnitude of the population accessing abortion care, policy makers may fail to pass evidence-based

surveyhub@norc.org) for researchers who meet the criteria for access to confidential data.

**Funding:** This study was funded by a grant from a philanthropic foundation that makes grants anonymously. The funders had no role in study design, data collection and analysis, decision to publish, or preparation of the manuscript.

**Competing interests:** The authors have declared that no competing interests exist.

abortion policy and may under-allocate resources, thereby failing to meet the needs of the population.

In recent years, several methods have been developed to better estimate the prevalence of items related to sensitive behaviors and topics such as abortion [8, 10, 11]. One method that is newer to abortion research is the list experiment (also known as the item count technique) [10, 11]. The list experiment historically has been used in political science surveys to elicit prevalence of opinions that are socially unacceptable or that respondents would otherwise prefer to conceal, such as racism and sexism [10]. For example, the 2008–2009 American National Election Studies (ANES) panel study included a list experiment, which listed five items including the item of interest, which related to a Black person becoming president, and then asked respondents to report the number of the items that they disliked–not which ones, just the number of total items that they disliked. More specifically, when using the list experiment technique, the sample is randomly split into two or more groups of respondents, and then each group is asked to report the number of items in a list that apply to them. Each group is given 1) a control list of non-sensitive items only, or 2) the list of the same non-sensitive items with a sensitive item added (e.g., abortion utilization). Respondents are asked to report the total number of items that apply to them, rather than identifying the specific items individually. The difference in the mean number of items reported between the lists with and without the sensitive item is equivalent to the population-level prevalence estimate of the sensitive item. This original list experiment method requires a large sample size [8, 10, 11], and thus, a variation referred to as the "double list experiment" instead incorporates two pairs of lists (List 1 and List 2) (Fig 1). In the double list experiment, List 1 and List 2, each have two versions one with the sensitive item and one without the sensitive item (List 1A & List 1B; List 2A & 2B). Within a list, the non-sensitive items do not change (i.e., List 1A and List 1B have all the same non-sensitive items; the only things that varies is whether the list also contains the sensitive item or not). Half of the participants receive List 1A and List 2A while the other half of the participants receive List 1B and List 2B. Typically, half of the participants will receive a list set with the sensitive item included in the first list while the other half will see the sensitive item in their second list. The reported numbers across the groups are compared which yields two population-level estimates for the sensitive item (one for List 1 and one for List 2), which can then be averaged together or reported separately.

Given the intent of the list experiment design to increase reporting of sensitive or stigmatized experiences, the list experiment has been recently adopted in abortion research with varying degrees of success [12–23]. The indirect nature of measures like the list experiment are expected to yield a higher estimate of abortion utilization compared to direct questions, because there is greater privacy afforded to the respondent and their experiences are less visible to the researcher. Indeed, list experiment estimates of abortion have improved reporting over direct measures in several studies of abortion utilization in Iran, Liberia, Pakistan, Senegal, and the US [13–17, 22, 23]. However, the list experiment has not always succeeded in reducing underreporting. Several applications of the list experiment to measure abortion have failed to find higher estimates than direct questions, such as in studies conducted in Malawi, Turkey, Tanzania, and Vietnam [18–22]. Given the mixed performance of list experiments in measuring abortion across contexts, the method requires further study. Furthermore, the list experiment remains an important methodological tool even in locations such as the US that have some abortion surveillance data. First, official abortion statistics are incomplete for states that do not report to the Centers for Disease Control and Prevention, and no official statistics in the US include self-managed abortions, which might be expected to increase as clinic-based abortion care becomes increasing difficult to access. Additionally, if legal access to abortion is overturned in the US, or in some states, methods of collecting population-level estimates of

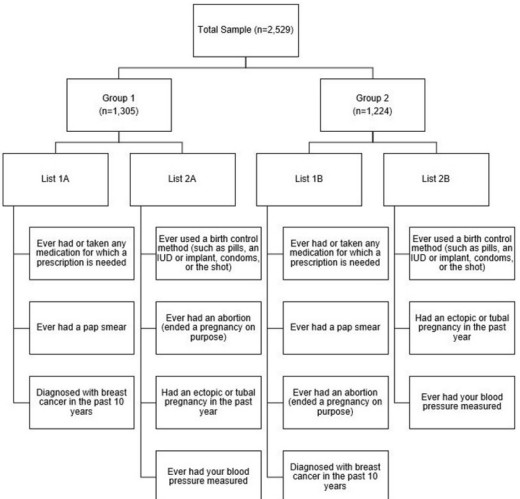

**Fig 1. Schematic of the double list experiment to measure abortion utilization in the Ohio Survey of Women (n = 2,529).**

abortion incidence could become increasingly relevant. Finally, scholars may wish to estimate abortion prevalence in sub-populations who other characteristics are not measured via the groups that compile national, state, or regional abortion prevalence data. To this end, we evaluated how an abortion-focused double list experiment performed relative to direct questions on abortion in a representative sample of adult reproductive-age women in Ohio, a state where abortion is legal, but where the state legislature has passed a large number restrictive laws in recent years [24–26]. Specifically, we examined whether the double list experiment would yield higher estimates of abortion utilization when compared to direct measures across the sample and within demographic subgroups.

## Methods & materials

### Data

We utilized data collected from October 2018 to June 2019 in the first wave of the Ohio Survey of Women, a statewide representative survey of 18- to 44-year-old adults who self-identified as a woman in the eligibility screening. Briefly, NORC at the University of Chicago used address-based sampling from the US Postal Services to create a sampling frame. NORC then stratified using area-level variables from the US Census Bureau. NORC oversampled individuals from rural Appalachian counties of Ohio to achieve an adequate sample size from this subpopulation. NORC sent letters to each household inviting them to participate with a weblink; nonrespondents were sent at least five follow-up invitations. Additional methodology from this survey is described elsewhere [27]. The response rate for the survey was 33.5% (n = 2529). Survey weights were used to ensure results were representative of the larger Ohio population.

### Direct abortion measures

We directly measured lifetime abortion utilization in two ways. First, all respondents were asked '*In your lifetime, have you ever been pregnant? Please include pregnancies that ended in miscarriage or abortion, in addition to births.*' Respondents who indicated that they never had a pregnancy that ended in a miscarriage, abortion or live birth were coded as "never utilized abortion services". Respondents who indicated they had ever been pregnant were then asked,

'*How many times have you had an abortion (ended a pregnancy on purpose)*?' Respondents could provide a number of abortions, indicate that they did not know, or that they preferred not to answer. With these two questions, we coded individuals as either having had an abortion or not.

For the second direct measure, all respondents were asked: '*Do you personally know someone such as a close friend, family member, or yourself, who has had an abortion? Please check all that apply*'. Respondents could select one or more of the following options: '*yes, a close friend*', '*yes, a family member*', '*yes, myself*', '*yes, someone else*', or '*no*'. Participants who selected '*yes, myself*' were coded as ever having had an abortion.

### Indirect measure

**The double list experiment.** To measure abortion history indirectly, we fielded a double list experiment to all participants. For the double-list experiment, two sets of lists were created (List 1 and List 2) (*Fig 1*). List 1 included three non-sensitive statements (*ever used or taken any medication for which a prescription is needed; ever had a pap smear; diagnosed with breast cancer in the past 10 years*) and *ever had an abortion (ended a pregnancy on purpose)*. List 2 also included three non-sensitive statements (*ever used a birth control method [such as: pills, an IUD or implant, condoms, or the shot], had an ectopic or tubal pregnancy in the past year; ever had your blood pressure measured*) and *ever had an abortion (ended a pregnancy on purpose)*. Two iterations of each list (A and B) were created for the two randomly selected sets of participants. List 1A contained only the non-sensitive items while List 1B contained the same non-sensitive items, with the addition of the statement about abortion. Conversely, List 2A contained all items (a different set of non-sensitive items plus abortion), while List 2B contained only the non-sensitive items from List 2A.

Respondents were randomly sorted into two groups, with one group receiving iteration A of Lists 1 and 2, and the other receiving iteration B. In this way, half of the respondents saw a set of lists with abortion on the second list while the other half of respondents saw a set of lists with abortion on the first list. We computed the difference in the results using survey weights between the two lists within a version of the list using the following equation:

$$\pi = 1/N_1 \Sigma Y_{T=1,i} - 1/N_0 \Sigma Y_{T=0,i}$$

where $\pi$ is the estimated population average, $N_1$ and $N_0$ are the population weighted number of respondents in who received the list with the sensitive statement and the list without the sensitive statement, respectively. $\Sigma Y_{t=1,i}$ and $\Sigma Y_{t=0,i}$ are the sums of the statements for each population-weighted $i$ respondent from the list with the sensitive statement and without the sensitive statement, respectively. We calculated the average of the two population estimates from the two list experiments to arrive at our final indirect estimate.

### Covariates

We examined abortion utilization by demographic characteristics, specifically age and socioeconomic status (SES). Other demographic characteristics could not be assessed due to small samples sizes in some of the categories. We categorized age as follows: 18–24, 25–29, 30–34, 35–39, and 40–44 years. We created a four-level categorical variable for socioeconomic status using two variables, household annual income and education: less than $75K and less than a college degree; $75K or more and less than a college degree; less than $75K and a college degree or higher; $75K or more and college degree or higher. Missing values in both education (2.6%) and income categories (14.7%) were imputed using hot-deck imputation. To minimize

the design effect of the survey, as recommended by NORC, we combined education and income to better fit the known education/income distribution in Ohio.

## Statistical analysis

We utilized descriptive statistics to compare the proportion of respondents who reported ever having an abortion in both direct measures and the double list experiment. We applied survey weights to all measures to estimate population proportions. For the direct measure, we described the proportion of all respondents who indicated they had an abortion, overall and stratified by the two covariates (age and SES). For the double list experiment, we evaluated the design effects using a likelihood ratio test ($\alpha = 0.05$) in the R package 'list.' We further evaluated the potential floor and ceiling effects by examining the percent of respondents who answered zero or three events in the control lists (lists without the abortion item i.e., 1A and 2B). We also examined demographic characteristics of the participants randomized to each list set to examine the extent to which randomization succeeded in balancing covariate distribution across groups. For the double list experiment, we similarly estimated the proportion of respondents who reported having an abortion by age and SES.

## Ethics approval statement

This was a secondary data analysis and was determined to be exempt for further review by The Ohio State University Institutional Review Board.

## Results

### Demographic characteristics

Approximately one quarter of our population were 18 to 24 years of age (24.7%) (*Table 1*), and nearly half (45.8%) had an income below $75,000 with less than a college degree. Just over half of the population ever had a pregnancy (53.2%). There were no major differences in sociodemographic characteristics between the two randomized groups (Lists 1A-2A and Lists 1B-2B).

### Assumption and validity checks

We observed no design effects for either list using the likelihood ratio test (List 1 p-value: 0.51; List 2 p-value: 0.87). In the control version of List 1 (A), 5.7% and 3.1% of the population reported experiencing no events and three (all) events respectively (*Table 2*). In the control version of List 2 (B), 4.6% and 2.8% of the population reported experiencing no events and three events, respectively. These low population percentages in the control lists indicate a low probability of floor and ceiling biases in the treatment lists.

### Estimates of abortion

Overall, 8.4% of the population indicated ever having had an abortion in response to the direction question about how many abortions they had experienced (*Table 1*). The second direct measure–based on whether the respondent identified themselves when asked if they knew someone who had an abortion–resulted in 8.0% of the population for ever-reporting an abortion. The double list experiment, asked to all participants, indicated that that 8.5% of population ever had an abortion. A little over 13% of the population did not answer either the first or the second part of the first direct abortion history question while 3.2% of the population did not answer the second direct abortion history question (*Table 1*). The missingness was generally lower in the lists compared with the direct questions, with missingness ranging from 1.7% to 4.1% in the population (*Table 2*).

**Table 1. Demographics characteristics of the Ohio Survey of Women, overall and by the two versions of the survey (n = 2,529).**

| Characteristics | Overall (n = 2,959) | | Group 1 [A] (n = 1,305) | | Group 2 [B] (n = 1,224) | |
|---|---|---|---|---|---|---|
| | n | Pop% | n | Pop % | n | Pop % |
| **Age** | | | | | | |
| 18–24 | 389 | 24.7 | 215 | 25.4 | 174 | 23.9 |
| 25–29 | 416 | 20.1 | 199 | 19.6 | 217 | 20.7 |
| 30–34 | 487 | 22.7 | 255 | 22.9 | 323 | 22.5 |
| 35–39 | 613 | 15.0 | 313 | 14.5 | 300 | 15.5 |
| 40–44 | 624 | 17.5 | 323 | 17.6 | 301 | 17.4 |
| **Socioeconomic Status [C]** | | | | | | |
| <$75K & <College degree | 947 | 45.8 | 465 | 44.1 | 482 | 47.7 |
| <$75K & College or higher | 537 | 11.1 | 281 | 11.3 | 256 | 11.0 |
| $75K+ & <College degree | 330 | 22.8 | 191 | 25.3 | 139 | 19.9 |
| $75K+ & College or higher | 715 | 20.3 | 368 | 19.3 | 347 | 21.4 |
| **Ever Pregnant** | | | | | | |
| Yes | 1477 | 53.2 | 752 | 52.2 | 725 | 54.4 |
| No | 839 | 37.0 | 446 | 37.2 | 393 | 36.7 |
| Don't know / Prefer not to answer | 118 | 5.2 | 52 | 4.6 | 66 | 5.8 |
| Missing | 95 | 4.6 | 55 | 6.0 | 40 | 3.1 |
| **Direct measure: Ever had an abortion?** | | | | | | |
| Yes | 192 | 8.4 | 91 | 8.3 | 101 | 8.6 |
| No | 2061 | 78.2 | 1074 | 77.9 | 987 | 78.6 |
| Missing | 276 | 13.4 | 140 | 13.8 | 136 | 12.8 |
| **Direct measure: Personally know someone who had an abortion?** | | | | | | |
| Yes, myself | 192 | 8.0 | 89 | 7.6 | 103 | 8.5 |
| No or yes, a close friend, family member or someone else | 2265 | 88.8 | 1180 | 89.1 | 1085 | 88.4 |
| Missing | 72 | 3.2 | 36 | 3.3 | 36 | 3.1 |
| **Indirect measure: Ever had an abortion** | --- | 8.5 | --- | 8.4 | --- | 8.5 |

[A] Group 1 received List 1A and List 2A

[B] Group 2 received List 1B and List 2B

[C] Household income.

In age-stratified analyses, estimates of abortion history increased with increasing respondent age (*Table 3*). Using the direct question, the youngest age group was least likely to report a history of abortion (18–24 years: 1.3%) while 35–39 year were most likely (13.7%-13.8%) and

**Table 2. Number of events selected by respondents between and across lists of the indirect measure of abortion utilization in the Ohio Survey of Women (n = 2529).**

| Number of Events | List 1 | | | | List 2 | | | |
|---|---|---|---|---|---|---|---|---|
| | Control (List 1A) (n = 1305) | | Experiment (List 1B) (n = 1224) | | Control (List 2B) (n = 1224) | | Experiment (List 2A) (n = 1305) | |
| | n | Pop % | n | Pop % | n | Pop % | n | Pop % |
| **0** | 50 | 5.7 | 57 | 7.1 | 40 | 4.6 | 44 | 5.5 |
| **1** | 171 | 16.6 | 162 | 16.1 | 176 | 14.8 | 180 | 15.1 |
| **2** | 1014 | 70.5 | 835 | 62.1 | 952 | 76.2 | 914 | 64.4 |
| **3** | 36 | 3.1 | 129 | 10.5 | 35 | 2.8 | 125 | 11.1 |
| **4** | | | 14 | 1.4 | | | 15 | 1.2 |
| **Missing** | 34 | 4.1 | 27 | 2.7 | 21 | 1.7 | 27 | 2.8 |

**Table 3. Precent of the population that report abortion utilization by direct and indirect measures and by age and socioeconomic status, the Ohio Survey of Women (n = 2529).**

| Characteristic | Direct Measure[1] | Direct Measure[2] | Indirect Measure[3,4] |
|---|---|---|---|
| | Pop % | Pop % | Pop % |
| Overall | 8.4% | 8.0% | 8.5% |
| Age | | | |
| 18–24 | 1.3% | 1.3% | --- |
| 25–29 | 7.5% | 7.1% | 11.1% |
| 30–34 | 8.2% | 8.2% | 9.2% |
| 35–39 | 13.7% | 13.8% | 17.4% |
| 40–44 | 15.3% | 13.3% | 23.3% |
| SES | | | |
| <$75K & < College | 9.9% | 9.6% | 12.8% |
| $75K+ & <College | 8.9% | 7.8% | 2.2% |
| <$75K & College+ | 5.3% | 5.4% | 7.9% |
| $75K+ & College+ | 6.3% | 6.1% | 4.3% |

[1] Direct measure used two questions: 1) the first asked about ever being pregnant (including miscarriage and abortion) and then asking abortion utilization among those reporting pregnancies.

[2] Direct measure asked to all participants about abortion utilization by friends, family, themselves, and others.

[3] Indirect measure used a double list experiment which averaged population estimates between the two lists.

[4] Valued among subgroups could take on negative values if sample size in a subgroup was not large enough or bias was present.

40–44 years (13.3%-15.3%). Similarly, in the list experiment, as age increased, abortion history also generally increased. In contrast to the direct measures, however, the lowest abortion history occurred in the 30–34 years (9.2%). Similar to the direct question, the oldest age group (40–44 years: 23.3%) had the highest estimate of abortion history; however, the estimate was much higher in the list experiment than what was observed with the direct questions. In the list experiment, an estimate of abortion utilization could not be obtained for the youngest age group (18–24 years) due the small number of events in this group. Comparing the direct measures and the double list experiment across SES, in all but two of the categories ($75K+ and less than a college degree; and $75K+ and a college degree or higher), the double list experiment yielded higher estimates of abortion utilization. In sum, the list experiment was inconsistent when providing estimates of history of abortion with some cases being higher than the direct questions and in some cases being lower than the direct questions.

Comparing the two lists in the double list experiment, we found large differences by age and SES subgroups (*Table 4*). For example, for age category 25–29 years, List 1 yielded an estimate of 17.4% while List 2 yielded an estimate of 4.9%. In the <$75K and a college degree or higher, List 1 yielded an estimate of 4.3%, while List 2 yielded an estimate of 11.6%. Additionally, some of the estimates resulted in negative percentages–suggesting that respondents reported a higher number of experiences for the control list–list without the sensitive item–than they did in response to the treatment list–list with the sensitive item indicating that the list set did not provide viable estimates.

## Discussion

Abortion is highly stigmatized in the US [2–7], making it likely that abortion occurrences are under-counted in direct questions [8]. In this representative sample of self-identified adult women of reproductive age in Ohio, we examined lifetime experience of abortion. Among the population, 8.4% and 8.0% reported ever having an abortion in response to the two direct

**Table 4. Percent of the population that reported abortion utilization using the indirect measure between and across the lists in the Ohio Survey of Women (n = 2529).**

| Characteristic | List 1 | List 2 | List Average |
|---|---|---|---|
| | Pop % | | Pop % |
| Overall | 8.4% | 8.5% | 8.5% |
| Age | | | |
| 18–24 | -16.9% | -5.1% | -11.0% |
| 25–29 | 17.4% | 4.9% | 11.1% |
| 30–34 | 4.9% | 13.5% | 9.2% |
| 35–39 | 19.4% | 15.4% | 17.4% |
| 40–44 | 26.1% | 20.4% | 23.3% |
| SES | | | |
| <$75K & < College | 12.1% | 13.6% | 12.8% |
| $75K+ & <College | 3.8% | 0.5% | 2.2% |
| <$75K & College+ | 4.3% | 11.6% | 7.9% |
| $75K+ & College+ | 4.9% | 3.6% | 4.3% |

measures, whereas 8.5% reported ever having an abortion when using the double list experiment. When findings are disaggregated by age and SES subgroups, in most instances, the double list experiment provided higher estimates of abortion than did direct measures. However, when the two lists were examined individually, they demonstrated wide variation by age and SES. Overall, the missingness was lower in the double list experiment compared to the direct questions.

The list experiment provided a marginally higher, but nearly identical, estimate of abortion prevalence when compared to the direct measure. However, given established underreporting of abortion in direct survey questions, we had hypothesized that list experiment estimates would be much higher. In six prior studies, list experiment estimates of abortion were substantially higher than those from direct questions; three of these studies were conducted in the US [13, 14, 23], one in Pakistan [15], one in Iran [16], and one Liberia [17]. All five used at minimum three control items, two of which would likely be prevalent in their population while the other statements were not prevalent in their population [13–17]. The rationale behind including high prevalent and low prevalent control items is to minimize the occurrence of floor and ceiling biases; however, this approach is only adopted when there are limited relevant control list items for inclusion. Mirroring the approaches of these other studies, we similarly included two high prevalence events and one low prevalence event in each of our control lists. Similar to other studies, we found no evidence to suggest a design effect, which occurs when respondents modify their answers because of the inclusion of abortion in the list [10, 11]. We intentionally selected all control items so that they would relate to reproductive health or normalized health events in order to minimize any design effect. Only one other list experiment on abortion, by Moseson et al., included related health behaviors [13], and all of the other studies in which the list experiment on abortion outperformed direct measures included normalized health items [14–17]. Future studies should adopt control lists that include related but non-stigmatized items for the control lists to help normalize the sensitive item and help prevent potential design effects.

Overall, the list experiment returned an estimate of lifetime abortion history nearly identical to that obtained from the direct questions. The design of the survey itself could have influenced these results. Given that the survey was self-administered and not administered by an interviewer, respondents could have felt less social desirability bias and therefore not

underreport their abortion history. However, the higher degree of missingness in the direct questions makes this idea unlikely. Additionally, the order of questions can influence responses. For example, the act of responding to the direct questions could cause respondents to guess at the purpose of the list experiment or feel compelled to provide consistent responses to the two types of questions. However, because the list experiment was conducted at the beginning of the survey and the direct questions were asked during the middle and end of the survey; it is unlikely that the direct questions influenced the list experiment. Among subgroup analyses, where sample size was underpowered from list experiment estimates, the list experiment estimate did not always return a higher estimate of abortion than the direct questions. There are several possible reasons for the unexpected results observed in subgroup analyses. The small sample size within the age and SES subgroups did not meet the recommended sample sizes for list experiments. Indeed, the need for large sample sizes with list experiments [8, 10, 11] is a resource-intensive limitation that motivated the innovation of the double list experiment in the first place. While we utilized the double list technique, the sample sizes among the subgroups remained low and underpowered for measurement of abortion experiences. Future studies measuring abortion utilization in specific subgroups should be planned a priori and use formal power calculations for each subgroup to ensure adequate sample size for these analyses.

In addition to sample size limitations, sub analyses by age may further have been biased by the use of age-related control list items (specifically: '*diagnosed with breast cancer in the past 10 years*' and '*had an ectopic or tubal pregnancy in the past year*'). The youngest respondents in the sample were 18 which means the control item asked about breast cancer risk related to their teenage years when the risk of this cancer is extremely low in comparison to the older age groups. If subgroups are of interest for future research, inclusion of control items equally applicable across these subgroups should be considered. Additionally, some control items such as ectopic and tubal pregnancy and pap smears may be poorly understood [13]. Consequently, some respondents might have misreported their experience of these control items. Therefore, when selecting non-sensitive items for the control lists, careful consideration should be given to how familiarity and understanding of all items, and their potential for misinterpretation.

We observed a lower prevalence of missing responses in the list experiment compared to the direct questions. One of the direct questions required respondents to answer two sequential questions which could have contributed to the higher reported missingness. In the other direct question, the missingness was slightly higher than what was reported for the list experiment. Given the small number of missing responses, it was not possible to fully investigate factors related to missingness; however, future studies could consider which factors influence missingness, and what gains, if any, vane be achieved by using one method or the other to improve reporting.

In short, as researchers continue to explore and evaluate the list experiment as a tool for measuring abortion, this application of the list experiment highlights important considerations for future iterations. Key features and lessons learned from this application of a double list experiment to measure abortion include: (1) the lower degree of missingness in the list experiment compared to direct questions, (2) the importance of a priori sample size planning for subgroup analyses of interest, (3) the careful selection of familiar/recognized control items that are related to the sensitive item of interest, and (4) selection of control items that are not differentially more or less likely based on any characteristics under analysis in subgroup analyses (i.e., age).

We recognize some limitations within this study. First, the abortion history obtained from the list experiment cannot provide additional details about timing, type, or other important details about the abortion experience–it only provides an estimate of abortion occurrence at

the population level. Second, both the direct and list experiment measures of abortion could have been interpreted broadly to include clinician-supported abortions as well as self-managed abortions; given the way the questions were asked, it is not possible to distinguish which abortion experience the participant reported. This limits the utility of the estimates to facilitate understanding of population-level demand for and experiences with self-managed versus facility-based abortion care. Third, one of the control list items asked about ectopic and tubal pregnancies. As the appropriate management of ectopic and tubal pregnancies is to remove the pregnancy, these are colloquially referred to abortions—as a result, some respondents may have been confused by this additional, if indirect, reference to abortion in List 2. Nonetheless, the randomization of the control and experimental lists would mean any bias introduced by this inclusion was non-differential. Furthermore, List 1 did not include this non-sensitive item and resulted in similar estimates which indicates this potential bias was minimal. Fourth, some of the non-sensitive statements were also asked about elsewhere in the survey. This may have raised concerns among some respondents that these questions could be used to identify their responses to the list experiment, which may have led to underreporting of abortion. However, this bias likely had a limited impact on our results as the list experiment is an average measure across the population and not all of the control items were included as individual survey questions. Fifth, only respondents who self-identified as women were eligible to participate in this study; as self-identified women are not the only individuals who require abortion care, our estimates do not capture the abortion experiences of transgender men, nonbinary people, and those of additional genders [28]. Finally, while our sample is representative of adult women of reproductive age in Ohio, it may not be generalizable to wider populations. Ohio is a large Midwestern state with population demographic characteristics similar to other Midwestern states [29], as well as a hostile political environment towards abortion [24–26] that is present in many other Midwestern and Southern states [1, 26]. However, these results may not be applicable to more diverse states nor ones that are less politically hostile towards abortion. Nevertheless, the sample being population-representative to adult reproductive-aged women in Ohio strengthens the generalizability of our findings and offers a unique lens on those most likely to utilize abortion services in the state.

## Conclusions

Underreporting of abortion is a persistent issue [8], impacting the capacity to make evidence-based policy decisions. The list experiment offers the opportunity to better measure abortion, but has some important limitations, including the large sample size necessary to power it, the careful design needed for selection of control items, and the exact wording of the abortion statement. Future studies should incorporate the lessons learned from this application of the list experiment, in an effort to improve reporting of abortion in state-level representative surveys.

## Acknowledgments

The authors thank NORC for conducting the Ohio Survey of Women.

## Author Contributions

**Conceptualization:** Robert B. Hood.

**Formal analysis:** Robert B. Hood, Heidi Moseson.

**Funding acquisition:** Alison H. Norris.

**Methodology:** Robert B. Hood, Heidi Moseson, Mikaela Smith, Payal Chakraborty, Alison H. Norris, Maria F. Gallo.

**Writing – original draft:** Robert B. Hood.

**Writing – review & editing:** Robert B. Hood, Heidi Moseson, Mikaela Smith, Payal Chakraborty, Alison H. Norris, Maria F. Gallo.

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
