## [Decision Letter · Decision Letter 0]

11 Feb 2022

PONE-D-22-00167Comparison of abortion incidence estimates derived from direct survey questions versus the list experiment among women in OhioPLOS ONE

Dear Dr. Gallo,

Thank you for submitting your manuscript to PLOS ONE. After careful consideration, we feel that it has merit but does not fully meet PLOS ONE’s publication criteria as it currently stands. Therefore, we invite you to submit a revised version of the manuscript that addresses the points raised during the review process.

We look forward to receiving your revised manuscript.

Kind regards,

Flávia Bulegon Pilecco

Academic Editor

PLOS ONE

Journal Requirements:

2.Please review your reference list to ensure that it is complete and correct. If you have cited papers that have been retracted, please include the rationale for doing so in the manuscript text, or remove these references and replace them with relevant current references. Any changes to the reference list should be mentioned in the rebuttal letter that accompanies your revised manuscript. If you need to cite a retracted article, indicate the article’s retracted status in the References list and also include a citation and full reference for the retraction notice.

4. Thank you for stating in your financial disclosure:  

"This study was funded by a grant from a philanthropic foundation that makes grants anonymously. The funders had no role in study design, data collection and analysis, decision to publish, or preparation of the manuscript."

PLOS ONE requires you to include in your manuscript further information about the funder so that any relevant competing interests can be assessed. Please respond to the following questions:

a. Please state whether any of the research costs or authors' salaries were funded, in whole or in part, by a tobacco company (our policy on tobacco funding is at http://journals.plos.org/plosone/s/disclosure-of-funding-sources)  

b. Please state whether the donor has any competing interests in relation to this work (see http://journals.plos.org/plosone/s/competing-interests) . 

c. Please state whether the identity of the donor might be considered relevant to editors or reviewers’ assessment of the validity of the work.

d. If the donors have no perceived or actual competing interests, please state: “The authors are not aware of any competing interests”. 

This information should be included in your cover letter. We will amend your financial disclosure and competing interests on your behalf.

Reviewers' comments:

Reviewer's Responses to Questions

**Comments to the Author**

1. Is the manuscript technically sound, and do the data support the conclusions?

Reviewer #1: Yes

Reviewer #2: Yes

2. Has the statistical analysis been performed appropriately and rigorously? 

Reviewer #1: Yes

Reviewer #2: Yes

3. Have the authors made all data underlying the findings in their manuscript fully available?

Reviewer #1: No

Reviewer #2: Yes

4. Is the manuscript presented in an intelligible fashion and written in standard English?

Reviewer #1: Yes

Reviewer #2: Yes

5. Review Comments to the Author

Reviewer #1: The manuscript "Comparison of abortion incidence estimates derived from direct survey questions versus the list experiment among women in Ohio" presents the results of a survey asking a direct as well as a list experiment question on abortion in a context, where abortions are stigmatized. The results show that in this context the list experiment did not improve reporting compared to a direct question. I think the manuscript is well written and analysed, and the topic is of interest. I have made some suggestions on how to further improve the manuscript, but many of them are rather minor.

1) I think the principle of double list experiment could be explained more clearly in the introduction section. In particular, it is not clear whether the non-sensitive control items are the same or different in between the two lists or the two randomised groups. While it is explained later in the methods section, it would be good to already mention it here.

2) While the list experiment did not yield less underreporting, the authors state that "missingness was generally lower in the lists compared with the direct questions". Could this perhaps present a small advantage of the list experiment compared to a direct question? Yet, this may not be enough to convince researchers to use the list experiment, given that a direct question is easier to link to respondents' individual-level characteristics. I think a short discussion on this topic, perhaps including some information about the characteristics of the respondents who responded to one but not the other type of question, would be of interest.

3) Was the direct questions or the list experiment asked first in the survey questionnaire? I am wondering if the women felt they could not truthfully answer the list question if they had already misreported their abortion experiences in the direct questions. In addition, given that there were two direct questions about abortion, I am wondering if the respondents who felt unconfortable sharing their abortion experience, guessed what the list experience was aiming to get at and chose (again) not to disclose their abortion? A discussion on whether a different study design (e.g. one where there is no direct question or one where it is only asked after the list experiment) may yield better results would be interesting.

4) The results presented in the Supplemental Table 1 seem important, as they suggest that perhaps the control items included in the lists were not ideal and could thus perhaps have led to the dissappointing results vis-a-vis the list experiment. I would suggest showing these results in the main text and not in the appendix.

5) While I found this article methodologically very interesting and worth publishing, I think it would be useuful if the authors discussed the rationale for using a list experiment in a context like the US. Given that the results can only be used to calculate an estimate of abortion occurrence at the population level, and that such an estimate already exists in the US from other sources, the authors should provide a rationale for aiming to perfect such a method in this context. Usually, this method is of the most interest in contexts, where an estimate of abortion occurrence in the population is not available from any source.

Minor comments:

INTRODUCTION

1st para: "normalize utilization of a common health experience"  how can one use a health experience? Perhaps could be expressed more clearly?

METHODS & MATERIALS

2nd para: "‘In your lifetime, have you ever been pregnant? Please include pregnancies that ended in miscarriage or abortion, in addition to births.’"  why were stillbirths not included in the list?

Why was 75k chosen as the cut-off for income? Does it represent a meaningful threshold of some kind? Could the authors elaborate, please? In the results section it looks like a large part of the population has an income below this threshold, so it seems a bit high.

Reviewer #2: Extremely important topic very carefully described and discussed. I recommend for publication. The comments below are minor points for the authors' considerations.

- In the abstract, the sentence: “When measured with the direct question, 8.4% and 8.0% of all respondents indicated ever having an abortion and with the list experiment, 8.5% indicated ever having an abortion.” It is a bit confusing what each reported % is in reference to, particularly because there are two percentages in reference to the direct questions, but it is a bit unclear what there are 2. This becomes clear in the results sections, but it might add clarity to slightly adjust how these numbers are presented here.

- Introduction: On the explanation regarding the experiment, I am not sure it is very clear to readers unfamiliar with list experiment. (ie two sets of lists vs on list pair, can be confusing). Perhaps making it a bit more clear that in the double list experiment there is a total of 4 lists, these are paired ( list A with and without the sensitive item and list B with and without the sensitive item) each responded will answer 2 lists, one from each pair and always one with and another without the sensitive item ( ie. List A without sensitive item and list B with sensitive item). This differs from the original count item method where respondents only answer 1 list, that might or might not have the sensitive item, and later these large groups are compared. Perhaps adding a reference to figure 1 here will also help.

- Methods & Materials: What does NORC stand for?

- Results:

. On “Demographic characteristics”, I would just specify that is household income.

. Would it be relevant to present Confidence intervals for some of the data presented?

. Regarding the lists comparisons: “Additionally, some of the estimates were negative percentages (a higher number of events in the control group than the treatment group) indicating that the list could not provide viable estimates.” It is unclear, what is considered the treatment group?

. I am not sure I follow the value of the lists comparisons as discussed and presented in the supplemental table 1 since they were answered by different groups. Are we comparing lists 1 and 2 here that were answered by the different groups? (Whoever answer list 1 A did not answer list 1B, correct?) So by “list 1” I assume is list 1A and 1B, that themselves differ by one item since they contain the exact items except for the sensitive item but were answered by different groups. Can we lump these different lists answered by different groups together and compare? Would this be a valid comparison? Can this data be disaggregated in this manner since now the groups are mixed?

6. PLOS authors have the option to publish the peer review history of their article (what does this mean?). If published, this will include your full peer review and any attached files.

Reviewer #1: No

Reviewer #2: No

---

## [Author Response · Author response to Decision Letter 0]

10 Mar 2022

Thank you for the opportunity to revise our manuscript for consideration by PLOS ONE. Please see our responses below to the peer reviewers. The reviewer comments were helpful, and we hope you agree that the revisions make the manuscript stronger.

Reviewer #1:

1. I think the principle of double list experiment could be explained more clearly in the introduction section. In particular, it is not clear whether the non-sensitive control items are the same or different in between the two lists or the two randomised groups. While it is explained later in the methods section, it would be good to already mention it here.

We agree with the reviewer and have added new text to the second paragraph of the Introduction section (p. 4). The revised text now reads (Lines 87-99):

“… the “double list experiment” instead incorporates two pairs of lists (List 1 and List 2) (Figure 1). In the double list experiment List 1 and List 2, each have two versions one with the sensitive item and one without the sensitive item (List 1A & List 1B; List 2A & List 2B). Within a list, the non-sensitive items do not change (i.e., List 1A and List 1B have all the same non-sensitive items; the only thing that varies is whether the list also contains the sensitive item or not). Half of the participants receive List 1A and List 2A while the other half of the participants receive List 1B and List 2B. Typically, half of the participants will receive a list set with the sensitive item included in the first list, while the other half will see the sensitive item in their second list. The reported numbers across the groups are compared which yields two population-level estimates for the sensitive item (one for List 1 and one for List 2), which can be averaged together or reported separately.”

2. While the list experiment did not yield less underreporting, the authors state that "missingness was generally lower in the lists compared with the direct questions". Could this perhaps present a small advantage of the list experiment compared to a direct question? Yet, this may not be enough to convince researchers to use the list experiment, given that a direct question is easier to link to respondents' individual-level characteristics. I think a short discussion on this topic, perhaps including some information about the characteristics of the respondents who responded to one but not the other type of question, would be of interest.

We agree with the reviewer that the missingness is an important aspect of our study. However, given the small amount of missingness we are unable to fully investigate factors related to missingness. We have included a new paragraph (fifth paragraph, p. 14) in the Discussion section on missingness, which notes the need for future studies to investigate these factors. The new text reads (Lines 316-323):

“We observed a lower prevalence of missing responses in the list experiment compared to the direct questions. One of the direct questions required respondents to answer two sequential questions which could have contributed to the higher reported missingness. In the other direct question, the missingness was slightly higher than what was reported for the list experiment. Given the small number of missing responses, it was not possible to fully investigate factors related to missingness; however, future studies could consider which factors influence missingness, and what gains, if any, can be achieved by using one method or the other to improve reporting.”

3. Was the direct questions or the list experiment asked first in the survey questionnaire? I am wondering if the women felt they could not truthfully answer the list question if they had already misreported their abortion experiences in the direct questions. In addition, given that there were two direct questions about abortion, I am wondering if the respondents who felt unconfortable sharing their abortion experience, guessed what the list experience was aiming to get at and chose (again) not to disclose their abortion? A discussion on whether a different study design (e.g. one where there is no direct question or one where it is only asked after the list experiment) may yield better results would be interesting.

In this study, the list experiment was conducted near the beginning of the survey and the direct abortion questions were not asked until the middle and end of the survey. Because of this ordering it is unlikely that the direct abortion questions influenced how participants responded to the list experiment. We do agree with the reviewer that if this ordering was reversed then the direct questions could have influenced the list experiment. We have included the following text in the Discussion to address this issue (fourth paragraph, p. 13, Lines 288-293):

“Additionally, the order of questions can influence responses. For example, the act of responding to the direct questions could cause respondents to guess at the purpose of the list experiment or feel compelled to provide consistent responses to the two types of questions. However, because the list experiment was conducted at the beginning of the survey and the direct questions were asked during the middle and end of the survey; it is unlikely that the direct questions influenced the list experiment.”

4. The results presented in the Supplemental Table 1 seem important, as they suggest that perhaps the control items included in the lists were not ideal and could thus perhaps have led to the dissappointing results vis-a-vis the list experiment. I would suggest showing these results in the main text and not in the appendix.

We have moved this table to the results section and renamed it Table 4.

5. While I found this article methodologically very interesting and worth publishing, I think it would be useuful if the authors discussed the rationale for using a list experiment in a context like the US. Given that the results can only be used to calculate an estimate of abortion occurrence at the population level, and that such an estimate already exists in the US from other sources, the authors should provide a rationale for aiming to perfect such a method in this context. Usually, this method is of the most interest in contexts, where an estimate of abortion occurrence in the population is not available from any source.

We agree with the reviewer that an optimal context to use the list experiment is when a population estimate for abortion history/utilization is unknown. We believe that additional work to better understand and improve the list experiment’s performance in the US context is important for three key reasons. First, scholars may be motivated to estimate the prevalence of abortion use in key sub-groups among whom they are collecting other data (e.g., people with substance use disorders) and for whom abortion prevalence cannot be estimated from other sources, because those other sources are not examining the sub-group characteristics. Secondly, the US does not have a complete abortion surveillance system; states report to the CDC on an optional basis, and not all states do this reporting (for example, California, which has an estimated 15% of all abortions in the US, does not report this information to the CDC). While non-federal systems (e.g., Guttmacher Institute’s 2017 Abortion Provider Census) do provide estimates, these are also incomplete. Third, abortions outside of the clinical setting (self-managed abortions) are currently undercounted in the US. These would increase in number if abortion were made illegal in the US or some states. The list experiment offers the opportunity to capture self-managed abortion; such utilization has previously been pilot tested.1 It is vital to improve this method in US contexts given the possibility that access to abortion could be entirely eliminated in some US states. There are an increasing number of state-level abortion restrictions and abortion bans (many of which are currently blocked by federal courts because they are unconstitutional under Roe v. Wade). However, the entire country is waiting for the US Supreme Court decision in Dobbs v. Jackson Women’s Health Organization which directly challenges and could overturn Roe v. Wade. Many states, including Ohio, are likely to ban abortion if Roe falls. (Ohio already has a 6-week ban that has passed but is blocked by federal courts. Ohio legislators have also proposed an abortion “trigger ban” [SB 123] and a total ban [HB 480] that mirrors the enforcement mechanism of Texas’ SB 8). If abortion access is eliminated in several states, then state health departments that currently collect and provide data on abortions to the public would cease collecting this information, resulting in the optimal use context the reviewer mentions. We have updated the text of the Introduction (third paragraph, p. 5) to reflect this additional rationale for conducting this study. The added text reads (Lines 111-120):

“Furthermore, the list experiment remains an important methodological tool even in locations such as the US that have some abortion surveillance data. First, official abortion statistics are incomplete for states that do not report to the Centers for Disease Control and Prevention, and no official statistics in the US include self-managed abortions, which might be expected to increase as clinic-based abortion care becomes increasingly difficult to access. Additionally, if legal access to abortion is overturned in the US, or in some states, methods of collecting population-level estimates of abortion incidence could become increasingly relevant. Finally, scholars may wish to estimate abortion prevalence in sub-populations whose other characteristics are not measured via the groups that compile national, state, or regional abortion prevalence data.”

1 Moseson H, Filippa S, Baum SE, Gerdts C, Grossman D. Reducing underreporting of stigmatized pregnancy outcomes: results from a mixed-methods study of self-managed abortion in Texas using the list-experiment method. BMC Womens Health, 2019;19(1):113.

6. INTRODUCTION, 1st para: "normalize utilization of a common health experience"  how can one use a health experience? Perhaps could be expressed more clearly?

We have edited this sentence in the Introduction (first paragraph, p. 3). The sentence now reads (Lines 64-66):

“Having valid estimates of lifetime cumulative incidence of abortion is important to normalize utilization of this common9 health service and inform the development of evidence-based polices to improve access to reproductive healthcare.”

7. METHODS AND MATERIALS, 2nd para: "‘In your lifetime, have you ever been pregnant? Please include pregnancies that ended in miscarriage or abortion, in addition to births.’"  why were stillbirths not included in the list?

We agree with the reviewer that the exclusion of stillbirths from this list is potentially ambiguous; while it was assumed that the term “births” would include both stillbirths and live births, this could have been made explicit to the respondents. However, we as the authors of this manuscript did not design the questionnaire and thus utilized the questions as asked.

8. Why was 75k chosen as the cut-off for income? Does it represent a meaningful threshold of some kind? Could the authors elaborate, please? In the results section it looks like a large part of the population has an income below this threshold, so it seems a bit high.

We agree with the reviewer that the threshold for income seems high. However, when conducting this survey NORC found that that income had a high degree of missingness (14.7%). Subsequently, they utilized hot-deck imputation to derive income values for participants who were missing these data. After checking their imputation and checking the weighting to Ohio demographics, NORC recommended utilizing a combined education and income variable with the following four categories (1) <$75K & <College degree; (2) ≥$75K & <College degree; (3) <$75K & ≥College degree; and (4) ≥$75K & ≥College degree. NORC found that this categorization reduced bias and the design effect based on raking performed on the post-imputed dataset to ensure the dataset was representative of the target population. We decided to utilize these same categories for these same reasons and to be consistent with studies previously published using these survey data.

Reviewer #2:

9. In the abstract, the sentence: “When measured with the direct question, 8.4% and 8.0% of all respondents indicated ever having an abortion and with the list experiment, 8.5% indicated ever having an abortion.” It is a bit confusing what each reported % is in reference to, particularly because there are two percentages in reference to the direct questions, but it is a bit unclear what there are 2. This becomes clear in the results sections, but it might add clarity to slightly adjust how these numbers are presented here.

We have added some clarifying language to the Abstract (first paragraph, p. 2) in two different places. First, we added (Lines 41-43):

“We aimed to evaluate whether a list experiment resulted in higher reporting of abortion experiences than did two direct questions.”

We also added (Lines 49-51):

“When measured with two different direct questions of abortion history, 8.4% and 8.0% of all respondents…”

10. Introduction: On the explanation regarding the experiment, I am not sure it is very clear to readers unfamiliar with list experiment. (ie two sets of lists vs on list pair, can be confusing). Perhaps making it a bit more clear that in the double list experiment there is a total of 4 lists, these are paired ( list A with and without the sensitive item and list B with and without the sensitive item) each responded will answer 2 lists, one from each pair and always one with and another without the sensitive item ( ie. List A without sensitive item and list B with sensitive item). This differs from the original count item method where respondents only answer 1 list, that might or might not have the sensitive item, and later these large groups are compared. Perhaps adding a reference to figure 1 here will also help.

We agree with the reviewer that this text is confusing and have updated the text in the Introduction (second paragraph, p. 4) and referenced Figure 1. The revised text now reads (Lines 87-99):

“… the “double list experiment” instead incorporates two pairs of lists (List 1 and List 2) (Figure 1). In the double list experiment List 1 and List 2, each have two versions one with the sensitive item and one without the sensitive item (List 1A & List 1B; List 2A & List 2B). Within a list, the non-sensitive items do not change (i.e., List 1A and List 1B have all the same non-sensitive items; the only thing that varies is whether the list also contains the sensitive item or not). Half of the participants receive List 1A and List2A while the other half of the participants receive List 1B and List 2B. Typically, half of the participants will receive a list set with the sensitive item included in the first list, while the other half will see the sensitive item in their second list. The reported numbers across the groups are compared which yields two population-level estimates for the sensitive item (one for List 1 and one for List 2), which can be averaged together or reported separately.”

11. Methods & Materials: What does NORC stand for?

NORC previously stood for the National Opinion Research Center at the University of Chicago. They have since dropped this name and instead use the acronym NORC as their official name. 

12. RESULTS, On “Demographic characteristics”, I would just specify that is household income.

We had added a footnote to Table 1 to indicate that this is household income.

13. Would it be relevant to present Confidence intervals for some of the data presented?

We had initially considered including confidence intervals with our analysis but opted not to include them for one key reason, being that our analysis was primarily descriptive in nature and thus confidence intervals were not necessary for a hypothesis generating study.

14. Regarding the lists comparisons: “Additionally, some of the estimates were negative percentages (a higher number of events in the control group than the treatment group) indicating that the list could not provide viable estimates.” It is unclear, what is considered the treatment group?

We agree with the reviewer that this verbiage is not clear. We have revised this sentence in the Results (fifth paragraph, p. 11); it now reads (Lines 247-250):

“Additionally, some of the estimates resulted in negative percentages – suggesting that respondents reported a higher number of experiences for the control list – the list without the sensitive time – than they did in response to the treatment list – the list with the sensitive item) indicating that the list set did not provide viable estimates.”

15. I am not sure I follow the value of the lists comparisons as discussed and presented in the supplemental table 1 since they were answered by different groups. Are we comparing lists 1 and 2 here that were answered by the different groups? (Whoever answer list 1 A did not answer list 1B, correct?) So by “list 1” I assume is list 1A and 1B, that themselves differ by one item since they contain the exact items except for the sensitive item but were answered by different groups. Can we lump these different lists answered by different groups together and compare? Would this be a valid comparison? Can this data be disaggregated in this manner since now the groups are mixed?

We are grateful to the reviewer for highlighting this lack of clarity. The reviewer is correct that List 1 has two versions: A and B, each with the same non-sensitive items. List 1A is asked to Group 1 and List 1B (includes the sensitive item) is asked to Group 2. If we are understanding the reviewer’s question correctly, the proposed comparison of the same list across groups is exactly what we did for the main analysis. From responses to List 1 across the two different sample groups, the population estimate of abortion history is obtained by comparing the average number events reported between the two groups. List 2 follows a similar pattern with version A (includes the sensitive item) being asked to Group 1 and version B being asked to Group 2. Because the groups are randomized this allows for comparison of events between the groups and for the population estimate. We believe including the supplemental table is important because it demonstrates that the list experiment requires a large sample size to allow for subgroup analysis as well as the need to have non-sensitive items appropriate for these subgroups. If we have misinterpreted the reviewer’s feedback, we defer to the editor. 

Sincerely,

Maria F. Gallo, PhD

---

## [Decision Letter · Decision Letter 1]

23 May 2022

Comparison of abortion incidence estimates derived from direct survey questions versus the list experiment among women in Ohio

PONE-D-22-00167R1

Dear Dr. Gallo,

We’re pleased to inform you that your manuscript has been judged scientifically suitable for publication and will be formally accepted for publication once it meets all outstanding technical requirements.

Kind regards,

Miquel Vall-llosera Camps

Senior Editor

PLOS ONE

Reviewers' comments:

Reviewer's Responses to Questions

**Comments to the Author**

1. If the authors have adequately addressed your comments raised in a previous round of review and you feel that this manuscript is now acceptable for publication, you may indicate that here to bypass the “Comments to the Author” section, enter your conflict of interest statement in the “Confidential to Editor” section, and submit your "Accept" recommendation.

Reviewer #1: All comments have been addressed

2. Is the manuscript technically sound, and do the data support the conclusions?

Reviewer #1: Yes

3. Has the statistical analysis been performed appropriately and rigorously? 

Reviewer #1: Yes

4. Have the authors made all data underlying the findings in their manuscript fully available?

Reviewer #1: No

5. Is the manuscript presented in an intelligible fashion and written in standard English?

Reviewer #1: Yes

6. Review Comments to the Author

Reviewer #1: The authors have addressed all my comments and I have no further comments. Congratulations on having written an interesting paper!

7. PLOS authors have the option to publish the peer review history of their article (what does this mean?). If published, this will include your full peer review and any attached files.

Reviewer #1: No

---

## [Editor Report · Acceptance letter]

26 May 2022

PONE-D-22-00167R1 

Comparison of abortion incidence estimates derived from direct survey questions versus the list experiment among women in Ohio 

Dear Dr. Gallo:

I'm pleased to inform you that your manuscript has been deemed suitable for publication in PLOS ONE. Congratulations! Your manuscript is now with our production department. 

Kind regards, 

on behalf of

Dr. Miquel Vall-llosera Camps 

Staff Editor

PLOS ONE